# Hi Robot: Open-Ended Instruction Following with Hierarchical Vision-Language-Action Models

Lucy Xiaoyang Shi [1 2]   Brian Ichter [1]   Michael Equi [1]   Liyiming Ke [1]   Karl Pertsch [1 2 3]   Quan Vuong [1]
James Tanner [1]   Anna Walling [1]   Haohuan Wang [1]   Niccolo Fusai [1]   Adrian Li-Bell [1]   Danny Driess [1]
Lachy Groom [1]   Sergey Levine [1 3]   Chelsea Finn [1 2]
https://www.pi.website/research/hirobot

## Abstract

Generalist robots that can perform a range of different tasks in open-world settings must be able to not only reason about the steps needed to accomplish their goals, but also process complex instructions, prompts, and even feedback during task execution. Intricate instructions (e.g., "Could you make me a vegetarian sandwich?" or "I don't like that one") require not just the ability to physically perform the individual steps, but the ability to situate complex commands and feedback in the physical world. In this work, we describe a system that uses vision-language models in a hierarchical structure, first reasoning over complex prompts and user feedback to deduce the most appropriate next step to fulfill the task, and then performing that step with low-level actions. In contrast to direct instruction following methods that can fulfill simple commands ("pick up the cup"), our system can reason through complex prompts and incorporate situated feedback during task execution ("that's not trash"). We evaluate our system across three robotic platforms, including single-arm, dual-arm, and dual-arm mobile robots, demonstrating its ability to handle tasks such as cleaning messy tables, making sandwiches, and grocery shopping.

## 1. Introduction

A defining feature of intelligence is its flexibility: people not only excel at complex tasks but also adapt to new situations, modify behaviors in real time, and respond to diverse inputs, corrections, and feedback. Achieving this kind of flexibility is essential for robots in open-ended, human-centric environments. For instance, consider a robot tasked with tidying up a table after a meal: instead of rigidly following a single predefined set of steps, the robot would need to interpret dynamic prompts like "only take away someone's dishes if they are done eating," respond to corrections like "leave it alone," and adapt when faced with unfamiliar challenges, such as a delicate object that requires special handling. This paper aims to advance robotic intelligence by enabling robots to interpret and act on diverse natural language commands, feedback, and corrections – a step towards creating agents that reason through tasks, integrate human feedback seamlessly, and operate with human-like adaptability. If we can enable a robot to process and engage with complex natural language interaction, we can unlock not only better instruction following, but also the ability for users to guide a robot through new tasks and correct the robot in real time.

Achieving this level of flexibility and steerability in robotic systems is challenging. While standard language-conditioned imitation learning can follow simple, atomic instructions such as "pick up the coke can" (Brohan et al., 2022), real-world tasks are rarely so straightforward. Imagine a more realistic prompt, such as: "Could you make me a vegetarian sandwich? I'd prefer it without tomatoes. Also, if you have ham or roast beef, could you make a separate sandwich with one of those for my friend?" This requires not only understanding the language, but also the ability to situate commands within the current context and compose existing skills (e.g., picking up the roast beef) to solve a new task. If the robot further receives corrections and feedback ("that's not how you do it, you have to get lower, otherwise you'll keep missing"), these must also be integrated dynamically into task execution. This challenge resembles the distinction between Kahneman's "System 1" and "System 2" cognitive processes (Kahneman, 2011). The "automatic" System 1 corresponds to a policy capable of executing straightforward commands by triggering pre-learned skills, while the more deliberative

[1]Physical Intelligence [2]Stanford University [3]University of California, Berkeley. Correspondence to: Physical Intelligence <research@physicalintelligence.company>.

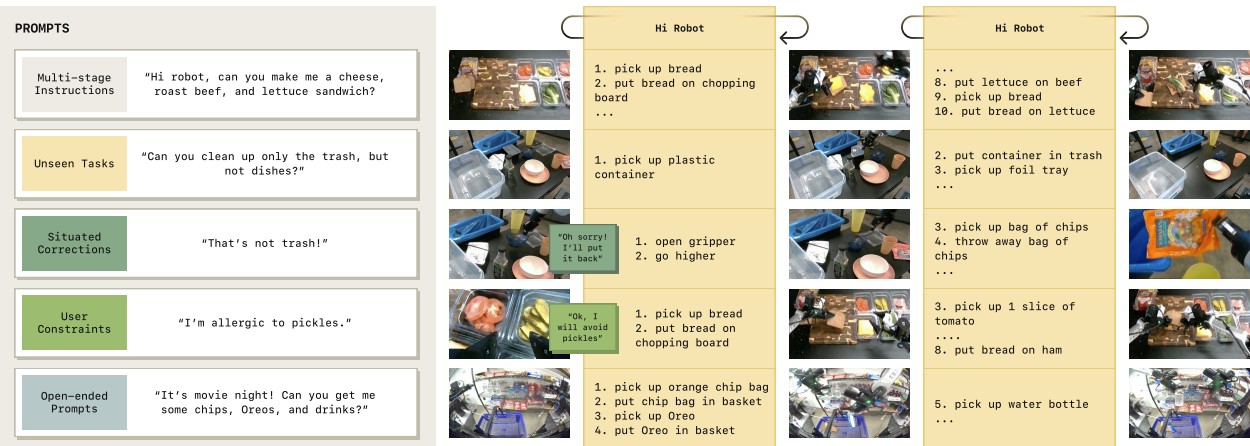

Figure 1: **Open-ended instruction following.** Hi Robot enables robots to follow multi-stage instructions, adapt to real-time corrections and constraints, complete unseen long-horizon tasks, and respond verbally when needed.

System 2 involves higher-level reasoning to parse complex long-horizon tasks, interpret feedback, and decide on an appropriate course of action. Prior work in robotic instruction following has largely focused on atomic instructions (Stepputtis et al., 2020; Jang et al., 2022; Brohan et al., 2022), addressing only System 1-level behaviors.

In this paper, we address the more intricate reasoning needed for complex prompts and feedback by introducing a hierarchical reasoning system for robotic control based on vision-language models (VLMs). In our system, the robot incorporates complex prompts and language feedback using a VLM, which is tasked with interpreting the current observations and user utterances, and generating suitable verbal responses and atomic commands (e.g., "grasp the cup") to pass into the low-level policy for execution. This low-level policy is itself a vision-language model finetuned for producing robotic actions, also known as a vision-language-action (VLA) model (Black et al., 2024; Brohan et al., 2023a; Kim et al., 2024; Wen et al., 2024). We expect that robot demonstrations annotated with atomic commands will not be sufficient for training the high-level model to follow complex, open-ended prompts, and we therefore need representative examples of complex prompt following. To acquire this data, we propose to *synthetically* label datasets consisting of robot observations and actions with hypothetical prompts and human interjections that might have been plausible for that situation. To this end, we provide a state-of-the-art vision-language model with a robot observation and target atomic command, and ask it to come up with a prompt or human interaction that may have preceded that observation and command, i.e. generating high-level policy prompts for different outcomes. By incorporating these synthetically-generated but situated examples into high-level policy training, our approach generalizes to diverse prompts and interjections while maintaining grounding in the robot's capabilities.

The main contribution of our paper is a **h**ierarchical **i**nteractive **robot** learning system (Hi Robot), a novel framework that uses VLMs for both high-level reasoning and low-level task execution. We show that our framework enables a robot to process much more complex prompts than prior end-to-end instruction following systems and incorporate feedback during task execution (Figure 1). While some of the individual components of this system, such as the low-level VLA policy, have been studied in prior work, the combination of these components along with our synthetic data generation scheme are novel and enable novel capabilities. We evaluate Hi Robot on diverse robots, including single-arm, dual-arm, and mobile platforms. Our evaluation requires the robots to perform a variety of tasks, including new combinations of skills seen during training, in the context of scenarios that span cleaning of messy tables, making sandwiches, and grocery shopping. Our experiments show that Hi Robot surpasses multiple prior approaches, including using API-based VLMs and flat VLA policies, in both alignment with human intent and task success. By grounding high-level reasoning in both verbal and physical interaction, Hi Robot paves the way for more intuitive and steerable human-robot symbiosis, advancing the potential for flexible intelligence in real-world applications.

## 2. Related Work

Our work relates to research on VLMs for robotic control, which we can categorize into two groups: directly training VLMs for robotic control and using VLMs out-of-the-box with pre-defined robot skills. In the former category, methods fine-tune VLMs to output robotic controls based on input images and language commands (Brohan et al., 2023a; Wen et al., 2024; Kim et al., 2024; Black et al., 2024; Liu

et al., 2024c; Li et al., 2024; O'Neill et al., 2024; Zawalski et al., 2024; Zheng et al., 2025; Pertsch et al., 2025) . While such methods have demonstrated impressive generalization and instruction-following, they are trained for relatively simple commands ("put the cup on the plate"). In contrast, we demonstrate tasks with intricate prompts and human interactions that require situated reasoning.

In the latter category, a number of methods use LLMs and VLMs to reason over robot observations and commands, and break up multi-stage tasks into simpler steps that can be performed by low-level controllers. Earlier methods of this sort used language models in combination with various learned or hand-designed skills (Huang et al., 2022; Brohan et al., 2023b; Liang et al., 2023; Shah et al., 2024; Singh et al., 2023; Wang et al., 2024; Li et al., 2025a), but such systems have limited ability to incorporate complex context, such as image observations, into the reasoning process. More recently, multiple works have use VLMs to output parameters for pre-defined robotic skills (Huang et al., 2023; Liu et al., 2024a; Nasiriany et al., 2024; Chen et al., 2024; Liu et al., 2024b; Stone et al., 2023; Qiu et al., 2024; Zhi et al., 2024). Such methods can process more complex commands and situate them in the context of visual observations, but these approaches have shown limited physical dexterity and limited ability to incorporate real-time language interaction with humans (with some exceptions discussed below). In contrast, our system utilizes VLMs for *both* high-level reasoning and low-level control, with a flexible language interface between the two. These design choices, along with a new synthetic data generation scheme, allow our system to achieve both significant physical dexterity and detailed promptability that prior works lack.

Many works aim to enable robotic language interaction with users, including model-based systems that parse language instructions and feedback and ground them via a symbolic representation of the scene (Swadzba et al., 2009; Matuszek et al., 2013; Namasivayam et al., 2023; Patki et al., 2019), and more recent learning-based methods that process feedback directly, typically with a hierarchical architecture (Liu et al., 2023; Xiao et al., 2024; Shi et al., 2024; Belkhale et al., 2024; Singh et al., 2024; McCallum et al.; Driess et al., 2023; Dai et al., 2024; Hu et al., 2023; Li et al., 2025b). Our work builds on the latter class of methods, where user feedback is incorporated via a high-level policy that provides atomic commands to a learned low-level policy. Unlike OLAF (Liu et al., 2023), which uses an LLM to modify robot trajectories, our approach can incorporate situated corrections based on the robot's observations, respond to those corrections in real time, and follow complex prompts describing dexterous manipulation tasks. While YAY Robot (Shi et al., 2024) can handle situated real-time corrections, it is limited to one prompt and

to the corrections seen in the human-written data; our approach leverages VLMs and a new data generation scheme to enable diverse prompts and open-ended corrections. Finally, RACER (Dai et al., 2024) can also incorporate situated corrections, but relies on a physics simulator to construct recovery behaviors; our approach only uses real robot demonstrations without intentional perturbations or corrections and is applicable to open-ended prompts.

# 3. Preliminaries and Problem Statement

A learned policy controls a robot by processing observation inputs, which we denote $\mathbf{o}_t$, and producing one or more actions $\mathbf{A}_t = [\mathbf{a}_t, \mathbf{a}_{t+1}, ..., \mathbf{a}_{t+H-1}]$, where we use $\mathbf{A}_t$ to denote an *action chunk* consisting of the next $H$ actions to execute (Zhao et al., 2023). Our system takes as input the images from multiple cameras $\mathbf{I}_t^1, ..., \mathbf{I}_t^n$, the robot's configuration (i.e., joint and gripper positions) $\mathbf{q}_t$, and a language prompt $\ell_t$. Thus, we have $\mathbf{o}_t = [\mathbf{I}_t^1, ..., \mathbf{I}_t^n, \ell_t, \mathbf{q}_t]$, and the policy represents the distribution $p(\mathbf{A}_t|\mathbf{o}_t)$. Prior works have proposed various methods for representing and training such policies (Zhao et al., 2023; Chi et al., 2023; Octo Model Team et al., 2024; Pertsch et al., 2025).

Since our focus will be specifically on complex, multi-stage tasks that require parsing intricate prompts and even dynamic user feedback, we need our policies to be able to interpret complex language and ground it via observations of the environment. A particularly powerful approach for handling such complex semantics is provided by vision-language-action (VLA) models (Black et al., 2024; Brohan et al., 2023a; Kim et al., 2024; Wen et al., 2024), which use vision-language model (VLM) pre-training to initialize the policy $p(\mathbf{A}_t|\mathbf{o}_t)$. A VLM is a language model that has also been trained to process image inputs, and represents a distribution $p(\ell'|\mathbf{I}, \ell)$ – the probability of a language *suffix* $\ell'$ (e.g., an answer to a question) in response to an image-language prefix consisting of an image $\mathbf{I}$ and a prompt $\ell$ (e.g., a visual question). The most commonly used VLMs represent $p(\ell'|\mathbf{I}, \ell)$ via an autoregressive decoder-only Transformer model, factorizing the distribution into a product of autoregressive token probabilities $p(\mathbf{x}_{t+1}|\mathbf{x}_1, ..., \mathbf{x}_t, \mathbf{I})$, where $\mathbf{x}_t$ denotes the $t^{\text{th}}$ token (not to be confused with a physical time step), and we have $\ell = [\mathbf{x}_1, ..., \mathbf{x}_{t_p}]$ and $\ell' = [\mathbf{x}_{t_p+1}, ..., \mathbf{x}_{t_p+t_s}]$, with $t_p$ the length of the prefix and $t_s$ the length of the suffix (Beyer et al., 2024). We also use such Transformer-based VLMs, but since we do not modify their architecture and their autoregressive structure is therefore not relevant to our discussion, we will use the more concise $p(\ell'|\mathbf{I}, \ell)$ notation to represent a standard VLM.

A standard VLA is produced by fine-tuning the VLM $p(\ell'|\mathbf{I}, \ell)$ such that the actions $\mathbf{A}_t$ are represented by tokens in the suffix $\ell'$, typically by tokenizing the actions via dis-

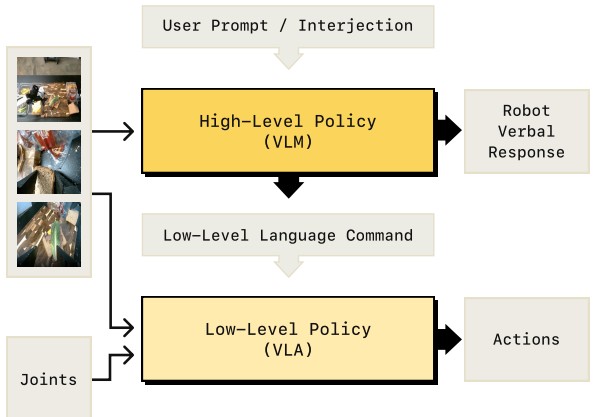

**Figure 2: Overview of hierarchical VLA.** The policy consists of a high-level and a low-level policy. The high-level policy processes open-ended instructions and images from base and wrist-mounted cameras to generate low-level language commands. The low-level policy uses these commands, images, and robot states to produce actions and optionally verbal responses.

cretization. We build on the $\pi_0$ VLA (Black et al., 2024), which additionally handles multiple images and continuous state observations $\mathbf{q}_t$, and modifies the VLM to output continuous action chunk distributions via flow-matching, but the high-level principles are similar. While such VLA models can follow a wide variety of language prompts (Brohan et al., 2023a), by themselves they are typically limited to simple and atomic commands, and do not handle the complex prompts and feedback that we study in this paper.

## 4. Hi Robot

We provide an overview of our method in Figure 2. Our approach decomposes the policy $p(\mathbf{A}_t|\mathbf{o}_t)$ into a low-level and high-level inference process, where the low-level policy consists of a VLA that produces the action chunk $\mathbf{A}_t$ in response to a simpler, low-level language command, and the high-level policy consists of a VLM that processes the open-ended task prompt, and outputs these low-level language commands for the low-level inference process. The two processes run at different rates: the low-level process produces action chunks at a high frequency, while the high-level process is invoked less often, either after a set time or upon receiving new language feedback. Thus, the high-level process essentially "talks" to the low-level process, breaking down complex prompts and interactions into bite-sized commands that can be converted into actions.

### 4.1. Hierarchical Inference with VLAs

Formally, the high-level policy $p^{\mathrm{hi}}(\hat{\ell}_t|\mathbf{I}_t^1, ..., \mathbf{I}_t^n, \ell_t)$ takes in the image observations and an open-ended prompt $\ell_t$, and produces an intermediate language command $\hat{\ell}_t$. The low-level policy $p^{\mathrm{lo}}(\mathbf{A}_t|\mathbf{I}_t^1, ..., \mathbf{I}_t^n, \hat{\ell}_t, \mathbf{q}_t)$ takes in the same type

of observation as the standard VLA described in Section 3, except that the language command $\ell_t$ is replaced by the output from the high-level policy $\hat{\ell}_t$. Thus, following the System 1/System 2 analogy, the job of the high-level policy is to take in the overall task prompt $\ell_t$ and accompanying context, in the form of images and user interactions, and translate it into a suitable task for the robot to do at this moment, represented by $\hat{\ell}_t$, that the low-level policy is likely to understand. Of course, for simple and familiar tasks, this is not necessary – if we simply want the robot to perform a task that the low-level policy was directly trained for, we could simply set $\hat{\ell}_t = \ell_t$ and proceed as in prior work (Brohan et al., 2022). The benefit of this hierarchical inference process is in situations where either the prompt $\ell_t$ is too complex for the low-level policy to parse, too unfamiliar in the context of the robot data, or involves intricate interactions with the user.

The high-level policy is represented by a VLM that uses the images and $\ell_t$ as the prefix, and produces $\hat{\ell}_t$ as the suffix. We describe how this model is trained in Section 4.3.

Since high-level inference is slower but also less sensitive to quick changes in the environment, we can comfortably run it at a lower frequency. A variety of strategies could be used to instantiate this, including intelligent strategies where the system detects when the command $\hat{\ell}_t$ has been completed before inferring the next suitable command. In our implementation, we found a very simple strategy to work well: we rerun high-level inference and recompute $\hat{\ell}_t$ either when one second has elapsed, or when a new interaction with the user takes place. This provides reactive behavior when the user provides feedback or corrections, while maintaining simplicity.

### 4.2. Incorporating User Interaction

The user can intervene at any point during policy execution and provide additional information and feedback, or even change the task entirely. In our prototype, these interventions take the form of text commands or spoken language (which is then transcribed into text). When the system receives a user intervention, the high-level inference is triggered immediately to recompute $\hat{\ell}_t$. The high-level policy has the option to include a verbal utterance $u_t$ in the command $\hat{\ell}_t$, which can be confirmations or clarifications from the robot. When $u_t$ is included, we use a text to speech system to play the utterance to the user, and remove it from $\hat{\ell}_t$ before passing it into the low-level policy.

When an interjection ("leave it alone") has been fulfilled, the user can signal to the robot that it may switch back to the previous command and continue the task execution. Notably, the responses of the high-level policy are *contextual*, because it observes not only the prompt $\ell_t$, but also the current image observations. Therefore, it can correctly

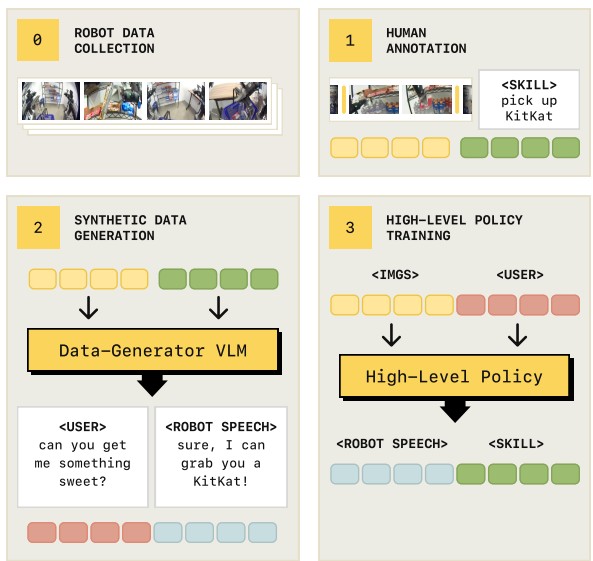

Figure 3: **Data collection and generation for training the high-level policy.** We first collect teleoperated robot demonstrations and segment them into short skills (e.g., *pick up KitKat*). Using this labeled data, we prompt a vision-language model (VLM) to generate synthetic user instructions (e.g., *"Can you get me something sweet?"*) and robot responses. The resulting dataset is used to train the high-level policy, which maps image observations and user commands to verbal responses and skill labels.

ground feedback like "that's not trash," which is not possible with language-only systems.

### 4.3. Data Collection and Training Hi Robot

To train Hi Robot in a scalable manner, we employ both human-labeled and synthetically generated interaction data, as illustrated in Figure 3. First, we collect robot demonstration data $\mathcal{D}_{demo}$ via teleoperation. This yields trajectories with coarse language annotations of the overall goal (e.g., *make a sandwich*). We then segment these full demonstration episodes into short skills, $\hat{\ell}_t$, such as *pick up one piece of lettuce*, which generally last between one and three seconds. We also heuristically extract basic movement primitives (e.g., small corrective motions) such as *move the right arm to the left* from the raw robot actions. The resulting dataset $\mathcal{D}_{labeled}$ contains a set of $(\hat{\ell}_t, \mathbf{I}_t^1, ..., \mathbf{I}_t^n)$ tuples that describe robot skills.

Next, we use a large vision-language model (VLM) $p^{gen}$ to produce synthetic user prompts and interjections $\ell_t$, and corresponding robot utterance $u_t$. Given $\mathcal{D}_{labeled}$, we prompt $p^{gen}$ with both the visual context $\mathbf{I}_t^1, ..., \mathbf{I}_t^n$ and the skill label $\hat{\ell}_t$ (e.g., *pick up the lettuce*). $p^{gen}$ then imagines an appropriate interaction that might have led to $\hat{\ell}_t$ in a real user interaction: it generates possible user prompts $\ell_t$ (e.g., *"Can you add some lettuce for me?"*) along with the robot's verbal responses and clarifications $u_t$. We detail the

generation of the synethetic dataset $\mathcal{D}_{syn}$ in Appendix A.

We train the high-level policy $p^{hi}(\hat{\ell}_t|\mathbf{I}_t^1, ..., \mathbf{I}_t^n, \ell_t)$ on $\mathcal{D}_{syn} \cup \mathcal{D}_{labeled}$ using the cross-entropy loss for next-token prediction. To train the low-level policy $p^{lo}(\mathbf{A}_t|\mathbf{I}_t^1, ..., \mathbf{I}_t^n, \hat{\ell}_t, \mathbf{q}_t)$, we use $\mathcal{D}_{labeled} \cup \mathcal{D}_{demo}$ using a flow-matching objective, following Black et al. (2024).

### 4.4. Model Architecture and Implementation

In our implementation, the low-level and high-level policies use the same base VLM as a starting point, namely the PaliGemma-3B VLM (Beyer et al., 2024). The low-level policy is the $\pi_0$ VLA (Black et al., 2024), which is trained by finetuning PaliGemma-3B with an additional flow matching "action expert" to produce continuous actions, while the high-level policy is fine-tuned on the image-language tuples described in Section 4.3 to predict commands. While we employ $\pi_0$ for our experiments, our framework is inherently modular, allowing for the integration of alternative language-conditioned policies as needed.

## 5. Experiments

In our experimental evaluation, we study a range of problems that combine challenging physical interactions with complex user interaction, including multi-stage instructions, live user feedback in the middle of the task, and prompts that describe novel task variations. We compare our full method to prior approaches and to alternative designs that use other high-level policy training methods. The aims of our experiments are:

1. Evaluate the ability of our method to follow a variety of complex textual prompts and live user feedback.
2. Compare our full method to prior approaches that train a flat instruction-following VLA policy or that use foundation models out-of-the-box for high-level reasoning.
3. Evaluate the importance of synthetic data and hierarchy for task performance and language following.

### 5.1. Tasks and Baseline Methods

We use three complex problem domains in our experiments, as shown in Figure 4.

**Table bussing** involves cleaning up a table, placing dishes and utensils into a bussing bin and trash items into the trash. The training data consists of full table cleaning episodes. This task is physically challenging because some items require nuanced grasping strategies (e.g., grasping a plate by the edge), the robot must pick up and singulate different objects, and in some cases might even manipulate some objects using others (e.g., picking up a plate with trash on it and tilting the plate to dump the trash into the trash bin). In our evaluation, the robot receives prompts that substan-

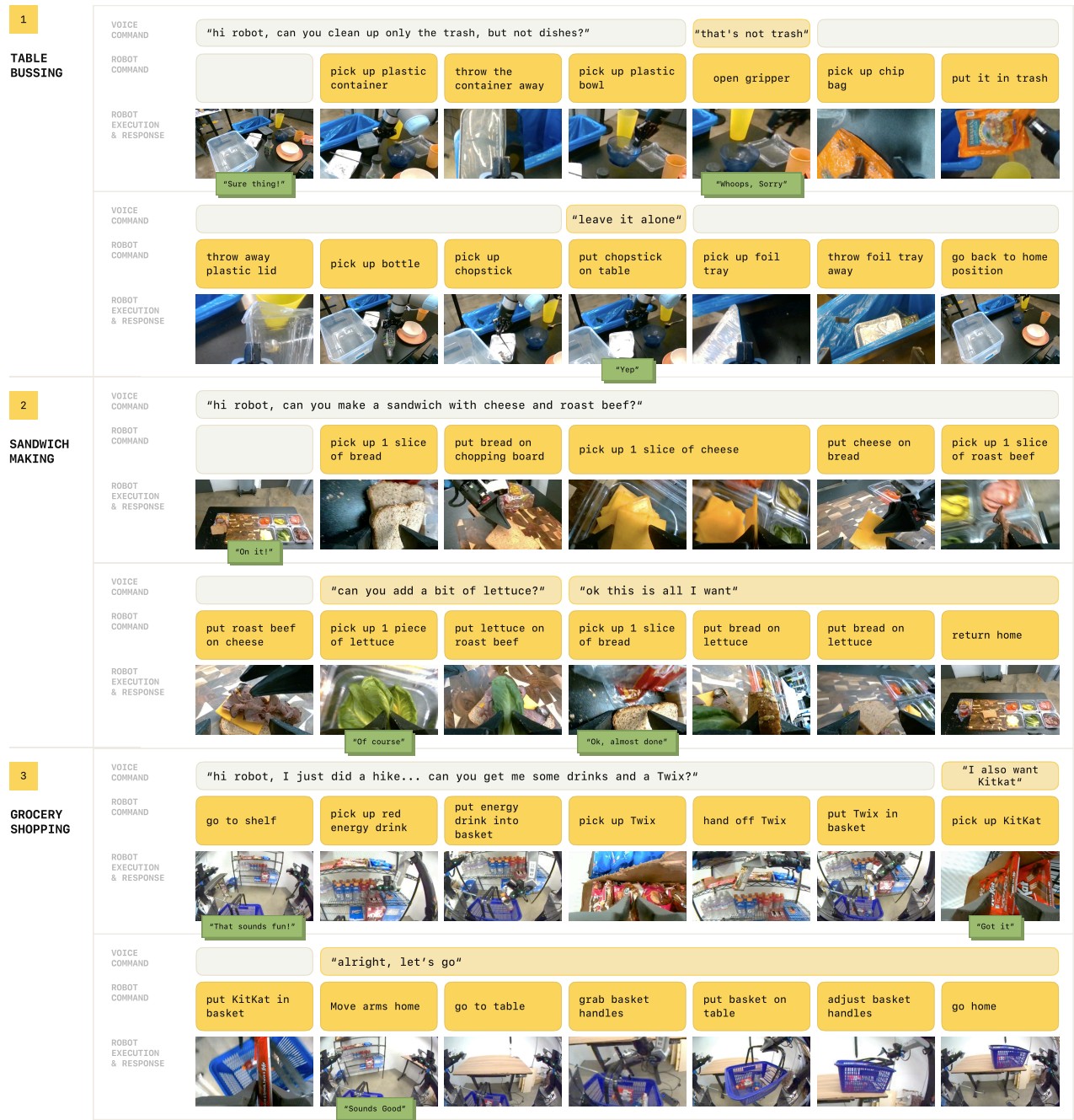

Figure 4: **Task domains used in our evaluation**. Across three domains, we evaluate complex instructions, intermediate feedback, and user interruptions. For example, in Table Bussing, when the user says, "that's not trash," the robot correctly puts the bowl back down instead of putting it away. All images are from policy rollouts.

tively alter the goal of the task, such as "can you clean up only the trash, but not dishes?", "can you clean up only the dishes, but not trash?", and "bus all the yellowish things". This requires the high-level model to reason about the task and each object (e.g., recognizing that reusable plastic cups are dishes, while paper cups are trash), then modify the

robot's "default" behavior of always putting away all items. This includes understanding what to do and also what *not* to do (e.g., avoid touching dishes when asked to collect only trash). The robot might also receive contextual feedback *during* the task, such as "this is not trash", "leave the rest", or "leave it alone," which require it to understand the

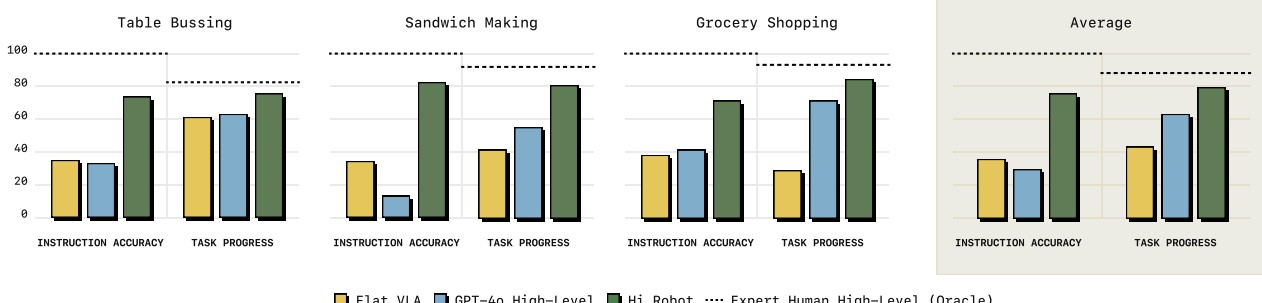

Figure 5: **Comparisons to Prior Methods.** Hi Robot outperforms GPT-4o and flat VLA on Table Bussing, Sandwich Making, and Grocery Shopping. Hi Robot averages over 40% higher instruction accuracy than GPT-4o, showing stronger alignment with user prompts and real-time observations, and approaches expert human guidance by leveraging its high-level policy.

interjection and respond accordingly.

**Sandwich making** requires the robot to make a sandwich, using up to six ingredients as well as bread. This task is physically difficult, because the robot has to manipulate deformable and delicate ingredients that have to be grasped carefully and placed precisely. The data contains examples of different types of sandwiches, with segment labels (e.g., "pick up one slice of bread"). We use this task to evaluate complex prompts, such as "hi robot, can you make me a sandwich with cheese, roast beef, and lettuce?" or "can you make me a vegetarian sandwich? I'm allergic to pickles", and live corrections, like "that's all, no more".

**Grocery shopping** entails picking up a combination of requested items from a grocery shelf, placing them into a basket, and placing the basket on a nearby table. This task requires controlling a bimanual mobile manipulator (see Figure 4) and interpreting nuanced semantics that involve variable numbers of objects. Examples of prompts include "hey robot, can you get me some chips? I'm preparing for a movie night", "can you get me something sweet?", "can you grab me something to drink?", "hey robot, can you get me some Twix and Skittles?", as well as interjections such as "I also want some Kitkat".

**Comparisons and ablations.** Our comparisons evaluate our full method and a number of alternative approaches, which either employ a different type of high-level strategy, or do not utilize a hierarchical structure. These include:

**Expert human high level**: This **oracle** baseline uses an expert human in place of the high-level model, who manually enters language commands for low-level behaviors that they believe are most likely to succeed at the task. This allows us to understand how much performance is limited by the low-level policy, with ideal high-level commands.

**GPT-4o high-level model**: This method uses the same high-level/low-level decomposition as Hi Robot, but queries the GPT-4o API-based model for the high level, while using the same low-level policy. GPT-4o is a significantly larger VLM than the one we use, but it is not

finetuned with our real and synthetic datasets. This comparison is similar to an advanced version of SayCan (Brohan et al., 2023b), which uses an out-of-the-box LLM as a high-level policy, while this baseline uses a VLM. To align GPT-4o with the robot's affordances, we carefully engineer the prompt to include task-relevant instructions that the low-level policy can follow, determined by ranking the most common skill labels in the human-annotated dataset, and ask GPT-4o to choose among them.

**Flat VLA**: This comparison directly uses the same $\pi_0$ low-level policy as in Hi Robot, but without any high level or synthetic data, representing a state-of-the-art approach for instruction following (Black et al., 2024).

**Flat VLA with synthetic data**: This ablation uses the $\pi_0$ low-level policy by itself, without a high-level model, but includes the synthetic data in the training data for the low-level policy, such that it can still process the complex prompts used in our evaluation. This baseline allows us to evaluate the benefit of hierarchy independent from the effect of synthetic data.

**Hi Robot without synthetic data**: This ablation corresponds to our method without synthetic training data, evaluating the importance of including diverse synthetically-generated prompts in training. This ablation can be seen as an advanced VLM-based version of YAY Robot (Shi et al., 2024), a prior system that uses a high-level model to predict language commands for a low-level model.

### 5.2. Metrics and Evaluation Protocol

We report two complementary metrics, measured by a human evaluator who is blind to the method being run. Each evaluation consists of 20 trials per task per method.

**Instruction Accuracy (IA).** This score measures how well the high-level policy's predicted instruction aligns with human intent, requiring multi-modal understanding of the current environment and prompt. If the prediction from the high-level model is consistent with both the user's command and the current observation, the evaluator marks it

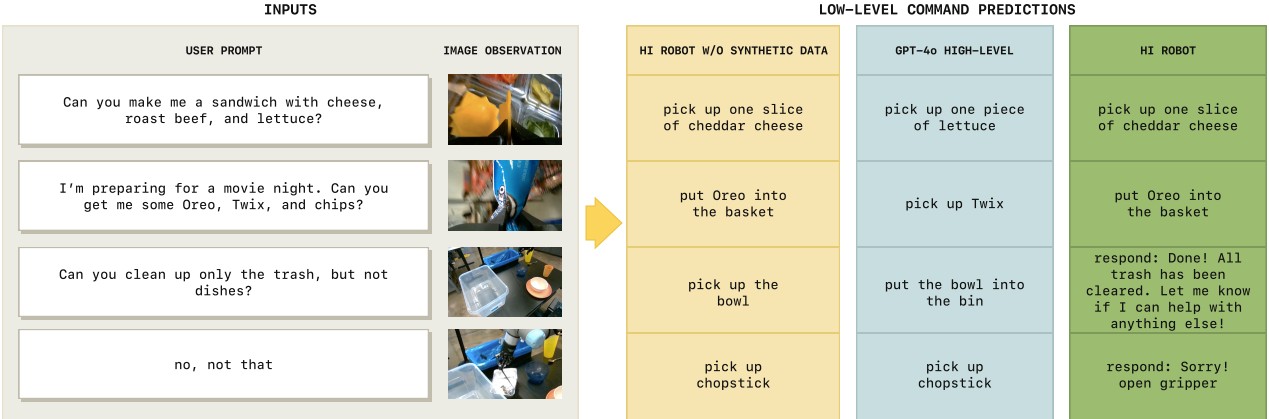

Figure 6: **Qualitative Command Comparisons.** GPT-4o often (a) misidentifies objects, (b) skips subtasks, or (c) ignores user intent. Hi Robot consistently produces commands aligned with the robot's ongoing actions and user requests. Without synthetic data, the high-level policy aligns well with image observations but ignores user constraints.

as a correct prediction; otherwise, it is labeled as incorrect. The Instruction Accuracy for a trial is then computed as the proportion of correct predictions out of the total number of predictions. For flat baselines, which lack interpretable language predictions, scoring is based on the evaluator's interpretation of the intent of the policy behavior.

**Task Progress (TP).** Since all tasks we evaluate are complex and long-horizon, we record task progress to provide a granular view of task completion. Task progress quantifies how closely the robot matches the intended goal and is computed by the proportion of objects that are successfully placed in their correct locations or configurations.

### 5.3. Core Results

We present results for our system and two key baselines: a GPT-4o policy and a flat VLA method. Quantitative and qualitative results are in Figure 5 and Figure 6, and we summarize our findings below.

**(1) Hi Robot excels at open-ended instruction following.** Across all tasks, Hi Robot exhibits substantially higher Instruction Accuracy and Task Progress, compared to GPT-4o and the flat baseline. It properly identifies, picks up, and places the correct items – even when prompted to handle only certain objects or omit ingredients (e.g., "I'm allergic to pickles"). In contrast, GPT-4o frequently loses context once physical interaction begins, issuing nonsensical commands (e.g., "pick up bermuda triangle") or sometimes labeling everything as "plate" or "spoon," which disrupts long-horizon planning.

**(2) Hi Robot shows strong situated reasoning and adaptation to feedback.** When users modify requests mid-task (e.g., "leave the rest," "I also want a KitKat"), Hi Robot updates low-level commands accordingly. GPT-4o, however, often fails to maintain a coherent internal state, leading to commands like picking up new objects when the gripper is still occupied or prematurely switching tasks. The flat baseline, on the other hand, does not react to real-time feedback.

**(3) Hi Robot is effective across diverse tasks, robots, and user constraints.** On single-arm, dual-arm, and mobile bimanual platforms, Hi Robot is able to handle distinct objects (from fragile cheese slices to tall bottles) while respecting dynamic constraints (e.g., "bus only yellowish items," "don't add tomatoes"). By contrast, the flat baseline and GPT-4o often revert to default behaviors (e.g., picking up every object in sight, or including almost all ingredients in a sandwich) when the prompt changes mid-episode.

**(4) Expert human guidance reveals the low-level policy's strengths but underscores the need for high-level reasoning.** With human high-level instructions, the low-level policy executes nearly flawlessly, showing that failures stem more from reasoning than actuation. However, solely relying on human input is not scalable. Hi Robot bridges this gap via a high-level VLM that aligns with user prompts and real-time observations, whereas GPT-4o's lack of physical grounding and the flat baseline's lack of high-level reasoning hinder performance.

### 5.4. Ablation Studies

We conduct two key ablations to isolate the contributions of (1) synthetic data for high-level reasoning, and (2) hierarchical decomposition vs. a single "flat" policy.

**(A) Synthetic data is critical for open-ended instruction following.** Comparing Hi Robot (trained on human-labeled + synthetic data) to a variant trained solely on human-labeled data shows that synthetic interactions significantly boost language flexibility (Figure 7). Without them, the ab-

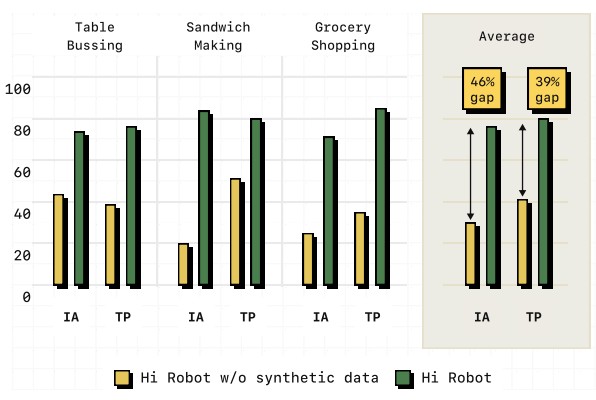

Figure 7: **Ablation on synthetic data.** Synthetic data is essential for handling open-ended instructions, as the model trained without it struggle with user-driven deviations, failing to integrate clarifications and constraints, whereas Hi Robot adapts seamlessly by leveraging diverse, compositional language prompts. (IA = Instruction Accuracy, TP = Task Progress)

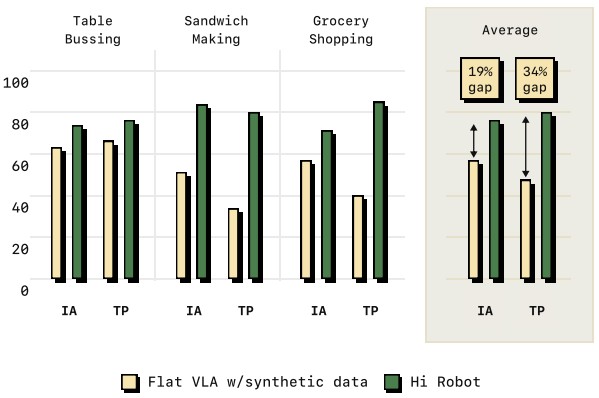

Figure 8: **Hierarchical policy vs. flat policy.** The hierarchical approach outperforms the flat variant trained on the same data, as it effectively integrates user feedback and partial instructions, whereas the flat model struggles with mid-task clarifications and nuanced task variations. (IA = Instruction Accuracy, TP = Task Progress)

lated model ignores clarifications (e.g., "this is not trash") or includes forbidden items (e.g., pickles), while Hi Robot smoothly adapts to such feedback, due to the broader coverage of compositional language in synthetic data.

**(B) Hierarchical structure outperforms a flat policy.** We next compare Hi Robot to a flat policy trained on the same synthetic data but without a separate reasoning step (Figure 8). The flat model often reverts to clearing all items or fails to handle partial instructions ("bus only the yellowish things"), whereas Hi Robot re-checks the prompt at each high-level step and responds coherently to mid-task updates. This suggests separating high-level reasoning from low-level control is benficial for multi-step coherence and adapting to dynamic user inputs.

## 6. Discussion and Future Work

We presented Hi Robot, a system that uses vision-language models (VLMs) in a hierarchical structure, first reasoning over complex prompts, user feedback, and language interaction to deduce the most appropriate next step to fulfill the task, and then performing that step by directly outputting low-level action commands. Our system can be thought of as a VLM-based instantiation of the "System 1" and "System 2" architecture (Kahneman, 2011). The deliberative "System 2" layer takes the form of a high-level VLM policy, which leverages semantic and visual knowledge from web-scale pre-training to reason through complex prompts and user interactions. The physical, reactive "System 1" layer also takes the form of a VLM, trained to directly output robot actions in response to simpler commands that describe atomic behaviors.

The two VLMs have nearly identical architectures, with the only difference being that the low-level policy uses flow matching to output the actions. Indeed, the separation of roles at the model level is not fundamental to this design: a natural step for future work is to combine both systems into one model, and draw the "System 1" vs "System 2" distinction purely at inference time. Future work could also interleave high-level and low-level processing more intricately – while our system simply runs high-level inference at a fixed but lower frequency, an adaptive system might simultaneously process inputs and language asynchronously at multiple different levels of abstraction, providing for a more flexible multi-level reasoning procedure.

Our system also has a number of limitations that could be studied in future work. While we show that our high-level policy can often break down complex commands into low-level steps that the robot can perform physically, the training process for this high level model relies in some amount of prompt engineering to produce synthetic training examples that induce this behavior. The training process decouples the high-level and low-level models, and they are not aware of one another's capabilities except through the training examples. Coupling these two layers more directly, e.g. by allowing the high-level policy to be more aware of how successfully the low-level policy completes each command, would be an exciting direction for future work. More generally, by instantiating both high-level and low-level reasoning via VLMs, we believe that this design opens the door for much more intricate integration of these components, such that future work might create robotic vision-language-action models that dynamically reason about inputs, feedback, and even their own capabilities to produce suitable situated response in complex open-world settings.

## Acknowledgments

We thank Ury Zhilinsky and Kevin Black for their help in setting up the data and training infrastructure. We thank Karol Hausman for valuable feedback and discussions on video demonstration and language-following evaluation. We are also grateful to Noah Brown, Szymon Jakubczak, Adnan Esmail, Tim Jones, Mohith Mothukuri, and Devin LeBlanc for their support in robot maintenance. We appreciate Suraj Nair and Laura Smith for their insightful discussions that helped with policy debugging. We also thank Claudio Guglieri for help in creating visualizations used in this paper and on the project website. Finally, we extend our deepest gratitude to the entire team of robot operators at Physical Intelligence for their immense contributions to data collection, annotation, and policy evaluations.

## Impact Statement

This paper presents work where the goal is to advance the fields of machine learning and robotics. There are many potential societal consequences of our work, none of which we feel must be specifically highlighted here.

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

# A. Synthetic Data Generation

## A.1. Scenario and Response Categorization

To ensure the quality and diversity of the synthetic data, we incorporate structured scenario classification and response categorization into the prompt design for $p^{gen}$, following (Stephan et al., 2024). Specifically, we classify interactions into different scenario types, such as *negative task* (where the user instructs the robot what *not* to do), *situated correction* (where the user adjusts an earlier command based on the evolving task state), and *specific constraint* (where the user specifies particular constraints, such as dietary preferences). In addition, we categorize the robot's responses into types such as *simple confirmations*, *clarifications*, and *error handling*. These classifications guide the generation process to ensure a broad range of user-robot interactions.

## A.2. Prompt Construction for Contextual Grounding

In prompt $\mathcal{P}$, we include a detailed description of the task (e.g., bussing a table, making a sandwich, grocery shopping) and instruct the model to ground responses in visual observations and prior context. A key advantage of leveraging large pretrained VLMs is their ability to incorporate world knowledge when generating interactions. For instance, the model can infer dietary constraints when generating prompts for sandwich-making, producing user commands such as "Can you make a sandwich for me? I'm lactose intolerant" and an appropriate robot response like "Sure, I won't put cheese on it." Similarly, it can reason over ambiguous or implicit requests, such as inferring that "I want something sweet" in a grocery shopping scenario should lead to suggestions like chocolate or candy.

To maintain consistency in multi-step tasks, we condition $p^{gen}$ on prior skill labels within an episode $\hat{\ell}_0, ..., \hat{\ell}_{t-1}$, allowing it to generate coherent user commands that account for past actions. For instance, if the robot has already placed lettuce and tomato on a sandwich, the generated user prompt might request additional ingredients that logically follow. This ensures that the synthetic interactions reflect realistic task progression rather than isolated commands. As such, we leverage $p^{gen}(\ell_t, u_t | \mathbf{I}_t^1, ..., \mathbf{I}_t^n, \hat{\ell}_0, ..., \hat{\ell}_{t-1}, \hat{\ell}_t, \mathcal{P})$ to produce a richer, more diverse synthetic dataset $\mathcal{D}_{syn}$ that provides meaningful supervision for training our high-level policy.

While in this work we generate a separate $\mathcal{D}_{syn}$ and train a separate high-level policy for each task (e.g., sandwich making vs. table cleaning) for clarity and ease of benchmarking, the architecture is readily amenable to a unified multi-task formulation. In principle, the same hierarchical approach could be used to train a single high-level policy across a multitude of tasks, facilitating knowledge transfer

between task domains and more robust, open-ended robot behavior.

# B. System and Robot Overview

Our system integrates speech-based interactions and real-time robotic control. Below, we detail the components of our system, including audio processing, GPU-based inference, and the robot configurations.

## B.1. Perception and Language Processing

For speech-based interaction, we use a consumer-grade lavalier microphone for audio input. Speech-to-text transcription is handled locally using Whisper large-v2 (Radford et al., 2023). For text-to-speech synthesis, we employ the Cartetia API to generate natural and expressive speech outputs.

## B.2. Inference Hardware

To support real-time inference, we utilize one to two NVIDIA GeForce RTX 4090 consumer-grade GPUs.

## B.3. Real-Time Inference Latency

We measured latency across components on consumer-grade RTX 4090.

**Low-Level Policy Per-Step Inference Times**

| Component | Time (ms) |
|---|---|
| Image encoding | 14 |
| Observation processing | 32 |
| Action prediction (x10) | 27 |
| Total (on-board) | 73 |
| Total (off-board + WiFi) | 86 |

**High-Level Policy (Single Decoding Step)**

- **RTX 4090**: 47 ms (prefill) + 13.2 ms (decode)
- **H100**: 17.3 ms (prefill) + 5.7 ms (decode)

These measurements confirm real-time feasibility at $\sim$10 Hz control rates. With action chunking (Zhao et al., 2023), we can use it to control robots at 50 Hz.

## B.4. Robot System Details

We employ three different robot configurations with various manipulation and mobility capabilities.

**UR5e.** This setup features a 6-DoF robotic arm equipped with a parallel jaw gripper. It includes two cameras: a wrist-mounted camera and an over-the-shoulder camera. The system operates within a 7-dimensional configuration and action space.

**Bimanual ARX.** This configuration consists of two 6-DoF ARX arms. The system is equipped with three cameras: two wrist-mounted cameras and one base camera. The combined system has a 14-dimensional configuration and action space, enabling dextrous bimanual manipulation tasks.

**Mobile ARX.** Built on the Mobile ALOHA (Fu et al., 2024) platform, this system integrates two 6-DoF ARX robotic arms mounted on a mobile base. The nonholonomic base introduces two additional action dimensions, resulting in a 14-dimensional configuration space and a 16-dimensional action space. Similar to the bimanual setup, it includes two wrist-mounted cameras and a base camera, providing robust visual feedback for navigation and manipulation.

## C. Model and Experiment Details

### C.1. Model Initialization

While our method can be trained from scratch or fine-tuned from any vision-language model (VLM) backbone, in practice we use PaliGemma (Beyer et al., 2024) as the base model. PaliGemma is an open-source, 3-billion-parameter VLM that offers a good balance between performance and computational efficiency. We unfreeze the full model for fine-tuning.

### C.2. Optimizer and Hyperparameters

We use the AdamW optimizer (Loshchilov & Hutter, 2017) with $\beta_1 = 0.9$, $\beta_2 = 0.95$, and no weight decay. Gradient norms are clipped to a maximum magnitude of 1. We maintain an exponential moving average (EMA) of the network weights with a decay factor of 0.999. The learning rate is warmed up over the first 1,000 steps and then held constant at $1 \times 10^{-5}$. We use a batch size of 512.

### C.3. Training Duration and Resources

Training the high-level policy is highly efficient, requiring approximately 2 hours on 8×H100 GPUs. The low-level policy follows a similar training pipeline, though training time may vary depending on the dataset size and the complexity of the target tasks for action prediction.

### C.4. Failure Cases

We observed the following failure modes:

1. **High-level**:
   - Difficulty with instructions requiring long-context reasoning, since the current system lacks memory

2. **Low-level**:

- Temporarily ignoring instructions: e.g., grabbing cheese when the robot is close to it despite user's lactose intolerance (due to training bias toward proximal objects)
- Error accumulation and out-of-distribution (OOD) recovery: e.g., dropped objects

Beyond the future directions discussed in the main text, several additional mitigations may help address observed limitations, including but not limited to:

- Stronger instruction-following model
- Long-context model
- Adversarial data generation for edge cases
- Diverse data collection including failure recovery and annotation

### C.5. System Prompt for GPT-4o

In the system prompt for GPT-4o, we include a description of the task, robot setup, and common instructions to select from. Below is an example for the Table Cleaning task.

Listing 1: GPT-4o Baseline Prompt

```
You are an AI assistant guiding a single-
arm robot to bus tables.

The robot can optionally place trash in the
 trash bin and utensils and dishes in the
plastic box.

Every 2 seconds, you can issue one
instruction from a provided list.
You will receive images from two cameras:
one for a global view and one on the robot'
 wrist for detailed views.

Interpret the user's instruction into one
from the provided list for the robot to
execute. Adhere strictly to the user's
instruction. If ambiguous, reason out the
best action for the robot. Only provide the
 exact instruction from the list without
explanation.

You will select your instruction from the
following list:
put food container in trash bin
pick up chopstick
drop wrapper in trash
pick up plastic plate
pick up the cup
pick up white bowl
place bowl to box
pick up spoon
place trash to trash bin
drop box in trash
place take out box to trash
move to the left
pick up container
drop plate in bin
pick up the trash
```

```
pick up plastic bowl
go higher
place spoon to box
pick up the paper container
drop fork in bin
pick up the bowl
pick up the plastic container
go lower
pick up box
move to the right
drop plastic lid into recycling bin
pick up wrapper
pick up the plate
put bowl in box
pick up the container
put the plate in the bin
pick up cup
put cup into box
throw it in the trash
pick up food container
pick up blue cup
drop the bowl into the bin
move towards me
pick up napkin
rotate counterclockwise
put the cup in the bin
throw trash away
rotate clockwise
drop plastic bowl into box
open gripper
pick up plastic cup
pick up the plate
close gripper
move away from me
go back to home position
```

