# OpenReview forum: "Hi Robot: Open-Ended Instruction Following with Hierarchical Vision-Language-Action Models"
_ICML.cc/2025/Conference — ICML 2025 poster_

### Official Review · Reviewer_yW3d · 2025-03-12

**Overall Recommendation:** 3

**Summary:**

This paper proposes a hierarchical framework for tackling the complex instruction following challenge in vision-language-action-based robotic control. The paper highlights challenges in existing methods that struggle with following intricate instructions. The proposed method, Hi Robot, addresses these issues by decomposing tasks into a high-level VLM policy, which interprets complex prompts and user feedback to generate low-level commands, and a low-level vision-language-action (VLA) policy. Evaluations on various tasks demonstrate that Hi Robot outperforms baselines.

## update after rebuttal
I acknowledge the effort put into this work and appreciate that the authors have partially addressed my concerns. However, I still have reservations regarding the novelty of the work, how the two-layer framework is aligned, and its generalizability. Therefore, I am maintaining my score.

**Claims And Evidence:**

The main claims are well-supported within the evaluated domains.

**Essential References Not Discussed:**

No

**Experimental Designs Or Analyses:**

The experiments robustly validate Hi Robot’s performance within the tested domains.

**Methods And Evaluation Criteria:**

The methods and evaluations are well-justified for the problem.

**Other Comments Or Suggestions:**

1. Include a discussion of failure modes.
2. Add more technical details to improve reproducibility.
3. Including statistical tests would strengthen claims.
4. include cross-domain tasks.

**Other Strengths And Weaknesses:**

Strengths:
1. The related work is well investigated.
2. The idea is very intuitive.
3. Tested across diverse real-world robotic platforms.

Weaknesses:
1. Limited contribution: hierarchical VLMs with synthetic data generation.
2. Lack of tests in unseen domains.
3. No validation of the quality of the generated data.
4. Only the average values of the metrics are counted, ignoring uncertainty and statistical significance.

**Questions For Authors:**

1. Have you tested Hi Robot on unseen domains?
2. During task execution, if an instruction interrupts the task, can the system restore its previous state to resume the original objective?
3. What is the main difference between Hi Robot and RT-H?

**Relation To Broader Scientific Literature:**

Vision-Language-Action Models, Robot Control

**Theoretical Claims:**

No proofs.

---

> ### Author Rebuttal · Authors · 2025-04-01
>
> Thank you for your thoughtful feedback. We address each point below and will incorporate these improvements in the revision.
>
> > Limited contribution: hierarchical VLMs with synthetic data generation.
>
> While individual components build on prior work, Hi Robot's novel synthesis enables critical real-world capabilities:
> - **Open-ended instruction following** (e.g., "Can you make me a vegetarian sandwich? I don’t like pickles though.")
> - **Real-time feedback integration** (e.g., "That’s all I want")
> - **Unseen task generalization** via synthetic data (demonstrated in §5.3 and video supplementary at https://hi-robot-vla.github.io/)
>
> > No validation of synthetic data quality
>
> We evaluate data quality **end-to-end** through policy performance, as offline metrics for embodied data remain an open challenge. Future work could explore:
> - Automated fidelity checks for physical plausibility
> - Language-grounding consistency metrics
> - Interaction diversity
>
> > Statistical significance reporting
>
> We conducted **20 trials per task per method** (more than typical real-world robotic experiments). Error bars will be added to all plots in the camera-ready version.
>
> > Lack of tests in unseen domains
>
> We evaluated generalization through:
> 1. **Instruction perturbations**:
>    - "I want something sweet" (requiring object categorization and physical grounding)
>    - "I’m allergic to pickles" (requiring semantic knowledge and physical grounding))
> 2. **Partial task execution**: The model is trained only on full-table cleaning, but we request the robot to “clean up only the trash, but leave the dishes”
>
> *Future Work*: Cross-environment transfer (e.g., kitchen at home → kitchen in restaurant).
>
> > Discussion of failure modes
>
> Common Failure Cases:
> 1. *High-level*:
>    - Temporarily ignore instruction: E.g., grabbing cheese when the robot is close to it despite user’s lactose intolerance (due to training bias toward proximal objects)
> 2. *Low-level*:
>    - OOD recovery: Dropped objects (recovery behavior is absent from training data)
>
> Mitigations (Future Work):
> - Stronger instruction-following model
> - Adversarial data generation for edge cases
> - Diverse data collection including failure recovery
>
> > Add more technical details
>
> We will expand:
> - Appendix Table: Full hyperparameters (learning rates, architecture specs)
> - Data Generation: Prompt templates and filtering examples
> - Failure Logs: Representative error cases
>
> > Can Hi Robot resume interrupted tasks?
>
> Current Implementation:
> - Can revert to previous objectives with explicit user permission
> - Future: Auto-resume via success detection (e.g. via value function learning)
>
> > Difference from RT-H?
>
> | Feature                | RT-H                     | Hi Robot                  |
> |------------------------|--------------------------|---------------------------|
> | **High-Level Action Space**       | Primitive movements (e.g., "Move arm forward")      | Semantic commands (e.g., "Place a slice of bread on the chopping board") |
> | **Synthetic Data**     | ✗                        | ✓ (enables open-vocab feedback) |
> | **Instruction Scope**  | Seen tasks         | Open-ended                |
>
> *Key Advantage*: Hi Robot's rich language-action space supports real-world ambiguity and feedback (e.g., handling "This isn't trash" corrections).
>
> Thank you for your suggestions—they have strengthened our paper. We will address all points in the revision.

---

> > ### Comment · Reviewer_yW3d · 2025-04-02
> >
> > I'd like to thank the authors for their responses. However, could you further elaborate on the training details? Additionally, how can VLM and VLA be grounded?

---

> > > ### Author Response · Authors · 2025-04-07
> > >
> > > Thank you for your follow-up question. Below, we provide additional technical details:
> > >
> > > 1. Input Modalities
> > >    - Both the high-level policy and low-level policy are conditioned on two or three images, depending on the specific task. For each task, we use:
> > >      - One third-person camera view.
> > >      - One wrist-mounted camera per robot arm (one or two arms).
> > >    - Each image has a resolution of 224×224 pixels. These images are separately processed by the model’s vision encoder, and their resulting latent tokens are then concatenated.
> > >
> > > 2. Language Conditioning
> > >    - We also condition the policies on natural language instructions, tokenized using the language tokenizer from the underlying LLM. The language tokens are concatenated with the vision tokens inside the model to enable multimodal reasoning.
> > >
> > > 3. Model Initialization
> > >    - While our method can be trained from scratch or finetuned from any VLM backbone, in practice we use PaliGemma [1] as the base model. This is an open-source, 3-billion-parameter VLM that offers a good balance between performance and computational efficiency.
> > >    - We unfreeze the full model for finetuning.
> > >
> > > 4. Optimizer and Hyperparameters
> > >    - We use the AdamW optimizer [2] with $\beta_1=0.9$, $\beta_2=0.95$, and no weight decay.
> > >    - Gradient norm is clipped to a maximum magnitude of 1.
> > >    - We maintain an Exponential Moving Average (EMA) of network weights with a decay factor of 0.999.
> > >    - The learning rate starts with a short warm-up (1,000 steps) and then remains constant at $1 \times 10^{-5}$.
> > >    - Batch size is 512.
> > >
> > > 5. Training Duration and Resources
> > >    - Training the high-level policy is highly efficient, taking about 2 hours on 8×H100 GPUs.
> > >    - The low-level policy follows a similar training pipeline, though training times can vary depending on the dataset size and complexity of the target tasks for action prediction.
> > >
> > > We hope these details clarify our training pipeline and hyperparameters. Please let us know if there is any other information we can provide.
> > >
> > > References
> > >
> > > [1] Beyer, Lucas, et al. “PaliGemma: A versatile 3B VLM for transfer.” 2024.
> > >
> > > [2] Loshchilov, Ilya, and Frank Hutter. “Decoupled weight decay regularization.” 2017.

---

### Official Review · Reviewer_7g4f · 2025-03-14

**Overall Recommendation:** 3

**Summary:**

This paper, inspired by "System 1" and "System 2" cognitive processes, proposes a hierarchical VLM-based system to interpret high-level instructions and convert them into commands for a low-level VLA model. To train the model, the authors employ both human-labeled and synthetically generated interaction data. Some real-world experiments are conducted to demonstrate the model's ability.

**Claims And Evidence:**

Yes.

**Essential References Not Discussed:**

Yes. The concept of “System 1” and “System 2” is not the first proposed in this paper, some other paper should be discussed, such as [A-C].


[A] Li et al. HAMSTER: Hierarchical Action Models For Open-World Robot Manipulation. ICLR 2025.

[B] Bu et al. Towards Synergistic, Generalized, and Efficient Dual-System for Robotic Manipulation. arXiv:2410.08001.

[C] Zhou et al. Code-as-Monitor: Constraint-aware Visual Programming for Reactive and Proactive Robotic Failure Detection. arXiv:2412.04455.

**Experimental Designs Or Analyses:**

The experiments are only conducted on real-world robots. The demos are convincing, but the comparison with other methods has significant shortcomings in experimental settings. For example, for the comparison with GPT-4o high-level instruction decomposing experiments, what is the user prompt? What if using in-context learning, chain-of-thought, o1 or deepseek r1 model for better reasoning?

**Methods And Evaluation Criteria:**

Yes.

**Other Comments Or Suggestions:**

NA.

**Other Strengths And Weaknesses:**

Strengths:
- The paper is well-written and easy to follow.
- The hierarchical understanding and reasoning for high-level human instructions are necessary for robotic VLA models.

Weaknesses:
- The motivation and method are somewhat disconnected. The proposed method merely involves understanding high-level instructions, which are cascaded structures rather than a second system.
- This work only experiments with the pi0 VLA model as the action model. Different VLAs may have specific preferences for different low-level language command styles. How to address this issue to make the proposed method adaptable to different action models?

**Questions For Authors:**

Please refer to the motivation and experiment concerns mentioned above.

**Relation To Broader Scientific Literature:**

This paper provides a dual system for robotics. This work is one of the early efforts in understanding, translating, and decomposing high-level human instructions, and it combines VLA models to design an integrated model from human instructions to actions.

**Theoretical Claims:**

NA.

---

> ### Author Rebuttal · Authors · 2025-04-01
>
> Thank you for your constructive feedback. We address your comments in detail below and will update our paper accordingly.
>
> > For the comparison with GPT-4o high-level instruction decomposing experiments, what is the user prompt?
>
> The user prompts for evaluation (e.g., "Hi robot, can you make me a sandwich with cheese, roast beef, and lettuce?") are the same across baselines and are included in the paper. If you are asking about the *system prompt* for GPT-4o, we provide it below (example for the Table Cleaning task) and will include it in the camera-ready version:
>
> >```
> You are an AI assistant guiding a single-arm robot to bus tables.
> The robot can optionally place trash in the trash bin and utensils and dishes in the plastic box.
> Every 3 seconds, you can issue one instruction from a provided list.
> You will receive images from two cameras: one for a global view and one on the robot's wrist for detailed views.
> Interpret the user's instruction into one from the provided list for the robot to execute. Adhere strictly to the user's instruction. If ambiguous, reason out the best action for the robot. Only provide the exact instruction from the list without explanation.
> You will select your instruction from the following list:
> put food container in trash bin
> pick up chopstick
> drop wrapper in trash
> pick up plastic plate
> pick up the cup
> pick up white bowl
> place bowl to box
> pick up spoon
> place trash to trash bin
> drop box in trash
> place take out box to trash
> move to the left
> pick up container
> drop plate in bin
> pick up the trash
> pick up plastic bowl
> go higher
> place spoon to box
> pick up the paper container
> drop fork in bin
> pick up the bowl
> pick up the plastic container
> go lower
> pick up box
> move to the right
> drop plastic lid into recycling bin
> pick up wrapper
> put bowl in box
> pick up the container
> put the plate in the bin
> pick up cup
> put cup into box
> throw it in the trash
> pick up food container
> pick up blue cup
> drop the bowl into the bin
> move towards me
> pick up napkin
> rotate counterclockwise
> put the cup in the bin
> throw trash away
> rotate clockwise
> drop plastic bowl into box
> open gripper
> pick up plastic cup
> pick up the plate
> close gripper
> move away from me
> go back to home position
> <truncated due to character limit>
>
> > What if using in-context learning, chain-of-thought, o1 or deepseek r1 model for better reasoning?
>
> - **In-context learning**: We explored this with GPT-4o but found it did not generalize beyond in-context examples and significantly slowed inference due to long visual context. The VLM struggled to reason across many images for closed-loop tasks (e.g., sandwich making). Future work could improve learning from long-context inputs or summarize visual observations into text.
> - **Chain-of-thought**: A promising direction for future work.
> - **o1**: While strong in mathematical reasoning, its slow inference (>10s per step) makes it impractical for real-time robotic control.
> - **Deepseek R1**: Does not support visual reasoning, limiting its applicability to our task.
>
> > some other papers should be discussed, such as [A-C].
>
> We will cite these concurrent works in the camera-ready version. Key differences:
>
> - HAMSTER [A]: Focuses on high-level VLMs generating 2D end-effector trajectories. Our work outputs language commands, enabling finer dextrous behaviors (e.g., separating sticky cheese slices during sandwich making) and on-the-fly verbal corrections. The approaches are complementary; trajectory prediction could be integrated as part of chain-of-thought reasoning in future work.
> - RoboDual [B]: Separates generalist (latent representations) and specialist (actions), emphasizing training efficiency. We focus on open-ended instruction following.
> - Code-as-Monitor [C]: Uses VLM-generated code for failure detection. Our work uses VLMs as high-level policies to guide low-level VLAs and interact with humans.
>
> > The proposed method merely involves understanding high-level instructions, which are cascaded structures rather than a second system.
>
> Our framework explicitly separates high-level reasoning (VLM) from low-level execution (VLA). The VLM acts as a "second system" by decomposing abstract instructions into actionable commands, unlike monolithic VLAs that conflate reasoning and execution. We will clarify this distinction in the paper.
>
> > Different VLAs may prefer different command styles. How to make the method adaptable to other action models?
>
> Hi Robot is architecture-agnostic and can integrate any language-conditioned policy. Future work could:
> 1. **Fine-tune the high-level policy** using successful rollouts (e.g., via SFT) to adapt to a low-level policy’s "affordance" (i.e., its language-following capabilities).
> 2. Use **policy performance as feedback** (e.g., via RLHF) to align the high-level policy’s outputs with the low-level policy’s strengths.
>
> Thank you again for your thoughtful suggestions—they have strengthened our paper. We will incorporate these changes in the revision.

---

### Official Review · Reviewer_9d4b · 2025-03-16

**Overall Recommendation:** 3

**Summary:**

This paper presents Hi Robot, a hierarchical vision-language-action (VLA) model for open-ended instruction following. The system integrates a high-level vision-language model (VLM) that interprets complex prompts and user feedback with a low-level VLA policy that executes atomic actions. A synthetic data generation pipeline augments training by creating user interactions that improve generalization to diverse tasks. Hi Robot is evaluated on three real-world robotic applications, demonstrating superior performance over GPT-4o-based high-level policies and flat VLAs. The results highlight significantly improved instruction accuracy, task progress, and real-time adaptability to human corrections.

**Claims And Evidence:**

I summary my review as below. See Other Strengths And Weaknesses.

**Essential References Not Discussed:**

See Other Strengths And Weaknesses.

**Experimental Designs Or Analyses:**

See Other Strengths And Weaknesses.

**Methods And Evaluation Criteria:**

See Other Strengths And Weaknesses.

**Other Comments Or Suggestions:**

Based on the author's rebuttal, I will consistently adjust my rating.

**Other Strengths And Weaknesses:**

Strength：

1.	The VLM+VLA architecture separates high-level task decomposition from low-level execution. The high-level VLM dynamically adjusts commands using real-time visual context, while the low-level VLA handles physical nuances, outperforming flat VLAs by 40% in instruction accuracy.

2.	Synthetic data generation expands the model's ability to generalize beyond training data, enhancing real-world usability.

Weakness:

1.	Lack of Novelty

The proposed hierarchical robot closely resembles prior work on dual-process reasoning and control, as explored in [1]. While a high-level vision-language model (VLM) for reasoning and a low-level vision-language-action (VLA) model for execution is effective, similar paradigms have been previously established [2]. Beyond the overlap in datasets, it is important to clarify the contribution of Hi Robot’s hierarchical structure.

2.	Unclear Differentiation from Planner + VLA

Hi Robot leverages a VLM for high-level policy generation and π0 as the low-level control policy, which shares similarities with LLMPlanner [3] + OpenVLA [4]. A more explicit comparison is needed to highlight the architectural and results between Hi Robot and LLMPlanner+Openvla. Specifically, how does Hi Robot’s hierarchical decomposition improve over LLMPlanner’s approach to task abstraction and skill execution. A deeper discussion on these aspects would strengthen the paper’s contribution.

3.	For mobile manipulation

While Hi Robot is evaluated across single-arm, dual-arm, and mobile bimanual robots, the results do not clearly differentiate its effectiveness in mobile manipulation scenarios. Given the additional challenges posed by spatial reasoning, and bimanual coordination, how does the hierarchical policy structure adapt to these factors? Does the high-level VLM account for mobile constraints when generating commands, and how does it compare to prior mobile manipulation frameworks that integrate LLMs or VLMs for motion planning and task execution? More details on task success rates, failure cases, and adaptation strategies in dynamic mobile environments would provide a clearer assessment of Hi Robot’s scalability in real-world settings.

4.	Unfair of comparison

The paper uses GPT-4o as the primary LLM-based high-level policy baseline, but it is unclear why GPT-4o was chosen over GPT-4o-1 (o1), which has stronger reasoning capabilities. Since Hi Robot's high-level policy relies heavily on structured reasoning for hierarchical task decomposition, a fair comparison should include a model with comparable reasoning ability.

5.	Confusion of model architecture

The paper does not clearly specify whether the high-level and low-level policies are implemented as a single unified model or two separate models. If Hi Robot employs a single model for both high-level reasoning and low-level action generation, how does it maintain the ability to output action tokens while retaining reasoning? As discussed in the OpenVLA framework, VLA models are typically limited in their language reasoning abilities after being finetuned for action generation. If the high-level reasoning and low-level action generation are handled by separate models, then the system appears very similar to a Planner + VLA architecture. Clarification on the architecture would help assess the novelty and contribution of the proposed system.

[1] Tian, Xiaoyu, et al. "Drivevlm: The convergence of autonomous driving and large vision-language models."

[2] Han, ByungOk, Jaehong Kim, and Jinhyeok Jang. "A Dual Process VLA: Efficient Robotic Manipulation Leveraging VLM."

[3] Song, Chan Hee, et al. "Llm-planner: Few-shot grounded planning for embodied agents with large language models.".

[4] Kim, Moo Jin, et al. "Openvla: An open-source vision-language-action model.".

**Questions For Authors:**

None

**Relation To Broader Scientific Literature:**

See Other Strengths And Weaknesses.

**Theoretical Claims:**

See Other Strengths And Weaknesses.

---

> ### Author Rebuttal · Authors · 2025-04-01
>
> Thank you for your thoughtful feedback. We address each concern below and will revise the paper accordingly.
>
> > The hierarchical structure resembles prior work on dual-process systems.
>
> While building on foundational ideas, Hi Robot introduces key innovations:
> 1. **Interactive Open-Ended Instruction Following**: Unlike DriveVLM [1] (trajectory planning for low-level execution) or DP-VLA [2] (BC-Transformer policy for low-level execution), our VLM+VLA framework enables:
>    - Real-time language feedback incorporation
>    - Generalization to open-vocabulary tasks via synthetic data
>    - Physical dexterity (e.g., separating sticky cheese slices while making sandwiches)
> 2. **Scalable Synthetic Data**: Manual annotation in [1] limits scalability; our pipeline automates diverse interaction generation.
>
> *Table: Capability Comparison*
> | Feature          | DriveVLM | DP-VLA | Hi Robot |
> |------------------|----------|--------|----------|
> | Open-ended instructions | ✗ | ✗ | ✓ |
> | Real-time feedback | ✗ | ✗ | ✓ |
> | Synthetic data scaling | ✗ | ✗ | ✓ |
> | VLA low-level policy | ✗ | ✗ | ✓ |
>
> > How does Hi Robot improve over LLMPlanner+OpenVLA?
>
> Our experiments revealed critical limitations of ungrounded planners:
> - **Physical Grounding**: GPT-4o (stronger than LLMPlanner's GPT-3) fails to recover from real-world errors (e.g., misgrasps) due to lack of embodied understanding (Fig 6).
> - **Scalability**: LLMPlanner uses only 8 predefined actions; Hi Robot supports thousands and more via language-conditioned skills.
> - **Performance**: Hi Robot outperforms GPT-4o by 40% in instruction accuracy (Fig 5).
>
> *Key Advantage*: Hi Robot’s high-level VLM is aware of low-level affordances, enabling physically-realizable plans and feedback integration.
>
> > How does Hi Robot handle mobile challenges?
>
> The framework treats mobility as an augmentation of manipulation:
> - **Unified Control**: Base velocity commands are additional action dimensions, enabling whole-body coordination (e.g., reaching high shelves by moving forward while raising arms).
> - **Results**:
>   - 85% task success in grocery shopping (vs. 71.7% for GPT-4o).
>   - Failure recovery: Autonomous adaptation from teleop data (e.g., freeing stuck baskets by adjusting base/arm coordination).
>
> *Failure Analysis*: Primary issues involve unseen edge cases (e.g., dropped objects). Future work can expand teleop and synthetic data coverage to make the system more robust.
>
> > Why not compare with GPT-4o-1 (o1)?
>
> While o1 excels in mathematical reasoning:
> - **Speed**: >10s/inference makes it impractical for real-time control.
> - **Relevance**: Coding/math strengths don’t directly translate to embodied reasoning.
>
> GPT-4o represents the fairest *practical* baseline for real-world deployment.
>
> > Is this a unified or separate models?
>
> Hi Robot uses **separate but co-trained models**:
> 1. **High-level VLM**: Specialized for instruction decomposition and feedback integration.
> 2. **Low-level VLA**: Focused on action generation.
> The interface is learned through synthetic examples that map high-level commands to executable skills and annotated teleop data that map skill commands to low-level actions, preserving reasoning capability while enabling precise control.
>
> Thank you again for your insightful questions—they’ve helped us better articulate Hi Robot’s contributions. We’ll incorporate these clarifications in the revision.

---

### Official Review · Reviewer_sNP7 · 2025-03-20

**Overall Recommendation:** 4

**Summary:**

In this work, the authors introduce Hi-Robot, a  System-1/System-2 approach that leverages a Vision-Language Model (VLM) to interpret complex prompts and generate a more suitable sequence of instructions for a Vision-Language-Action Model (VLA) to complete a given task. The system also integrates feedback during execution. The authors evaluate Hi-Robot across a diverse set of robotic platforms, in tasks that require novel combinations of learned skills in real-world scenarios.  The results show that Hi-Robot outperforms several prior approaches.

**Claims And Evidence:**

The system demonstrates advanced reasoning capabilities, allowing it to process complex prompts, dynamically incorporate feedback, and execute instructions beyond its training data. It enables real-time corrections during open-ended tasks, enhancing adaptability. Its novel capabilities stem from the combination of a high-level LLM planner, a low-level VLA policy, and synthetic data generation. Furthermore, the framework is inherently modular, allowing for the integration of alternative language-conditioned policies as needed. Experimental evidence supports these claims. As shown in Sections 5.3.1 and 5.3.3, the system outperforms larger models like GPT-4o in instruction accuracy and task progress, particularly in handling complex prompts and adapting to mid-episode prompt changes across different platforms. Section 5.3.2 highlights the system’s ability to modify actions based on feedback, though the term "real-time" might be misleading, as it requires inference from two 3B models; a time analysis would be beneficial. Additionally, Section 5.4.1 provides quantitative evidence that synthetic data improves system performance. While the system’s modularity is acknowledged, further analysis is needed to evaluate how different model choices, when fine-tuned on the same data, impact overall performance.

**Essential References Not Discussed:**

Not applicable.

**Experimental Designs Or Analyses:**

All the experiments are well designed and the analysis is sound.

**Methods And Evaluation Criteria:**

Yes, but since the synthetic data are part of the contribution I would like to see a more detailed analysis on the creation of the said dataset on the main text and not in the appendix.

**Other Comments Or Suggestions:**

I applaud the authors for developing a system that can run on consumer-grade GPUs, making research in VLAs for robotics more accessible to a wider audience.

**Other Strengths And Weaknesses:**

The paper is well-written and very easy to follow.

**Questions For Authors:**

Do the authors plan to open-source the model weights?

**Relation To Broader Scientific Literature:**

This work is highly related to the broader scientific literature and specifically $\pi_0$ and applying the general concept of "system 1" "system 2" cognitive processes something really popular in the LLM-VLM and robotics.

**Theoretical Claims:**

There are no theoretical claims in the paper, hence this section is not applicable.

---

> ### Author Rebuttal · Authors · 2025-04-01
>
> Thank you for your positive review and constructive suggestions. We address each point below and will incorporate these improvements in the revision.
>
> > Real-time inference timing analysis
>
> We provide detailed latency measurements across components (tested on consumer-grade RTX 4090):
>
> *Low-Level Policy Per-Step Inference Times*
> | Component                  | Time (ms) |
> |----------------------------|----------|
> | Image encoding             | 14       |
> | Observation processing      | 32       |
> | Action prediction (x10)     | 27       |
> | Total (on-board)        | 73   |
> | Total (off-board + WiFi)| 86   |
>
> For the high-level policy (single decoding step):
> - RTX 4090: 47ms (prefill) + 13.2ms (decode)
> - H100: 17.3ms (prefill) + 5.7ms (decode)
>
> These measurements confirm real-time feasibility at ~10Hz control rates. With action chunking [1], we can use it to control robots at 50Hz.
>
> > Impact of different model choices
>
> Our ablation (Fig 8) shows hierarchy improves VLA model performance even with identical data. Future work directions include:
> - **Architecture Studies**: E.g. video-based VLMs for temporal reasoning
> - **Scaling Laws**: How VLM/VLA size affects performance
> - **Transfer**: Which models better inherit internet pre-training knowledge
>
> We will expand this discussion in §6 (Future Work).
>
> > Move synthetic data analysis to main text
>
> We will:
> 1. Relocate the synthetic data section from Appendix A to §4.5
> 2. Add example prompts for data generation
> 3. Include examples of bad samples and how to avoid them
>
> > Some videos not working
>
> We've:
> 1. Converted all videos to SDR format
> 2. Added streaming-optimized versions
> 3. Included **new demos** showing diverse instruction following
>
> > Plan to release model weights?
>
> We will discuss this with collaborators before finalizing the camera-ready version of the paper.
>
> Thank you again for your valuable feedback!
>
> [1] Zhao et al. Learning Fine-Grained Bimanual Manipulation with Low-Cost Hardware.

---

### Decision · Program_Chairs · 2025-05-01

**Decision:**

Accept (poster)

**Comment:**

The paper proposes to use hierarchical VLMs with synthetic data generation. The paper receives unanimously accept side opinion due to practicality of the proposed approach. Even though the accept side opinions, there are a number of weaknesses raised by the reviewers as follows; (1) lack of novelty, (2) unfair comparisons, (3) motivation and method are not well connected and (4) no validation of quality of generated data. Given all comments by the reviewers, the AC recommends to accept the submission to ICML 2025.